# Developing Students Well-Being and Engagement in Higher Education during COVID-19—A Case Study of Web-Based Learning in Finland

**Minna Maunula** [1,*] , **Minna Maunumäki** [2], **João Marôco** [3] **and Heidi Harju-Luukkainen** [1]

1   Kokkola University Consortium Chydenius, University of Jyväskylä, 67701 Kokkola, Finland
2   Open University, University of Jyväskylä, 40014 Jyväskylä, Finland
3   William James Centre for Research, ISPA—Instituto Universitário, 1149-041 Lisbon, Portugal
*   Correspondence: minna.r.h.maunula@jyu.fi

**Abstract:** COVID-19 rapidly and extensively changed the normal everyday practices of societies, and there is no going back to the past. Universities also had to adapt and re-establish their normal routines, from policies to practices. In this article, we explore university students' experiences of web-based learning, their well-being, and engagement during the pandemic. As a theoretical framework, we use the concept of the university student engagement inventory (USEI), which includes behavioral, cognitive, and emotional dimensions. The data were collected during the COVID-19 pandemic from university students (N = 35) via an online survey and analyzed using a thematic content analysis. According to the results, university students experienced well-designed and pedagogically implemented web-based learning, teaching and guidance to enhance their own learning, well-being, and engagement in their studies. This suggests that web-based solutions for academic learning are justified but need to consider a range of well-being and engagement factors. What is still needed are innovative solutions that are pedagogically justifiable and consider the digital and human dimensions sustainably.

**Keywords:** web-based learning; well-being; engagement; university student

## 1. Introduction

COVID-19 has disrupted society's routines [1,2]. The pandemic created a state of emergency in almost all countries. As a result of this, many of the universities had to adapt and to recreate their normal routines from policies to practice, with little research evidence available [2]. For instance, Lemon et al. [3] highlighted the need to protect and maintain the wellbeing of staff and students, positioning the COVID-19 pandemic as a major catalyst of disruption [2,3]. Following the pandemic, web-based teaching and learning is becoming more widespread around the world and at all levels of education, including universities. Web-based teaching and learning can prove to be both an opportunity and a challenge for education providers, teachers, and students [4]. Web-based pedagogical solutions require special attention in both instructional design and implementation [5,6]. There is no difference in the effectiveness or depth of well-designed web-based learning compared to a traditional physically distributed learning situation [7,8]. Digitalization and web-based education has expanded, there is even talk of the datafication of education [9]. Web-based learning experiences can present challenges, as presented in research by Karkar-Esperat [10], who showed that some web-based learners faced challenges due to language skills, isolation, lack of guidance experience, and lack of motivation. In web-based learning, disengagement [11] and the superficial sharing of fragments of knowledge on online platforms, such as MOOCs are feared to impoverish learning processes [12]. Students also need to understand the reasons for using educational technology and learn how to use it

appropriately [11]. Following the COVID-19 pandemic, the digitalization of university education accelerated and occurred in a partly uncontrolled way, which brought new themes of pedagogical well-being [13], and issues of web-mediation and the internationalization [4] of pedagogical debate. It is important to also reflect on the expansive digitalization of education critically [9], but also to actively seek the benefits and opportunities of digitalization and web-based education. A variety of online high-impact practices prepare students for the demanding global and societal contexts of the future [14], but what is essential and fundamental, is students' well-being and engagement in web-based learning.

The shift to web-based learning is a challenge in terms of students' well-being and engagement [15], and it has been seen to diminish students' scholarship and study [16]. Student engagement is built in a social context, in different learning spaces, with peers, academic staff, and others. At the beginning of the pandemic, in online learning spaces, this was especially difficult since both the academic staff and students were learning how to interact in this new context. Students' well-being has been globally monitored during and after the pandemic, and there is more research evidence available on the effects of the pandemic on student engagement [1,17–19]. However, it is important to note that one of the factors that has been reported as alarming across the academic contexts has been related to student engagement. In one of these studies, three out of every four sampled students reported that low engagement during lectures hurt their online learning experience [18]. Engagement is therefore a critical component of online learning, according to these and others [17,18,20]. Especially, when moving towards a larger number of online and hybrid courses in an academic context, there is a growing need for research evidence on student engagement during online learning.

The notion of academic engagement is multidimensional and therefore difficult to define [21,22]. Academic engagement has been studied with multiple international scales and from varying perspectives [21]. It can be said to be a determinant of learning and academic success in a university context. According to Marôco et al. [23] student engagement is connected to physical and psychological energy during educational experiences. High engagement relates to positive individual and social well-being and high energy, and reduces dropout rates. It is connected to a lower risk of burnout [23] and to student self-efficacy [24,25] among others. According to Martin [25], engaged students feel a sense of purpose, demonstrate persistence, resilience, and emotional connection to others in learning spaces, feeling a sense of belonging and high self-efficacy. Therefore, according to Assunção et al. [26], student engagement can be viewed as a malleable, developing, and multidimensional construct that evolves over time in an individual self-determination. It is also important to note that it can be affected by interventions that enhance positive performance and prevent potential dropout [24,26].

In this paper, as our theoretical and analysis frame, we use the university student engagement inventory (USEI) model, developed by Marôco et al. [21,22]. The model has been previously used only in quantitative research and multiple measurements have been conducted with it in Europe, North and South America, Africa, and Asia [22,23,25–27]. The USEI includes university students' behavioral, cognitive, and emotional dimensions of academic engagement. The behavioral dimension is related to positive normative class behaviors (e.g., respecting the social and institutional rules). The cognitive dimension refers to students' thoughts, perceptions, and strategies related to the acquisition of knowledge or development of competencies to academic activities (e.g., learning approaches). The emotional dimension refers to positive and negative feelings and emotions related to the learning process, class activities, peers, and teachers [22,28,29]. Therefore, in this model, student engagement is conceptualized as a three-factor construct. From these premises, a research question was formulated for this study. How do university students experience web-based learning, their well-being, and engagement within the three-dimensional engagement construct. This was carried out with the help of an online survey with open-ended questions, involving 35 students during the COVID-19 pandemic. The data of this study is therefore textual and a theory driven thematic content analysis was applied. This

paper will support our understanding of student engagement but also, for the first-time, test Marôco et al.'s [22] university student engagement inventory (USEI) model's applicability to qualitative data.

## 2. Data

This study explores university students' experiences of web-based learning, well-being, and engagement. The research question is: what are students' experiences of web-based learning, well-being, and engagement during the COVID-19 pandemic? The data were collected through a Webropol survey in spring 2021, during the COVID-19 pandemic.

The survey included open-ended questions and was answered by 35 university students in Finland. Students responded to the Webropol survey anonymously and without identifying information. Students' answers to the open questions were comprehensive and varied in content. The interpretive approach of the study was based on a hermeneutic approach. According to hermeneutics, reality is complex and the meanings that construct it are individual and context-bound [30]. An attempt was made to describe and interpret the responses and individual meanings to explore the phenomenon under study, university students' experiences of online learning, well-being and engagement, and the relationship between them.

## 3. Methods

To answer the research question, the data were analyzed using an abductive thematic content analysis. This refers to a method of analysis that allows us to draw reproducible and valid inferences from texts to their contexts of use [30]. The hermeneutic analysis process began with an exploration of the data and discussions within the research team [30]. The data were then thematized according to the USEI-model of student engagement approach into three main themes: behavioral, cognitive, and emotional dimensions [22]. We refined the analysis and interpreted the main themes as positive and negative experiences of university students' experiences of web-based learning, well-being, and engagement. The analysis was a hermeneutic abductive dialogue between researchers, theory, and data [30]. As a result of the abductive reasoning, an overall understanding of university students' experiences of web-based learning, well-being, and engagement during the COVID-19 pandemic was formed. The results of the study are presented according to the main substantive themes: the behavioral, cognitive, and emotional dimensions, employing the USEI theoretical model of student engagement for abductive reasoning.

## 4. Results

The results section is constructed in order to reflect university students' experiences of web-based learning and wellbeing within the structure of the university student engagement inventory USEI model: the behavioral, cognitive, and emotional dimensions of university students' academic engagement.

### 4.1. The Behavioral Dimension

University students' experiences of web-based learning were mainly positive. According to the students, web-based learning emphasized student-centeredness, which was demonstrated in the pedagogical design of the studies. This was reflected in the possibility to personalize one's studies, to choose the study methods that suit one's needs, to set individual timetables, and to start studies flexibly during the academic year. The pedagogical basis of web-based learning was perceived by students as strengthening their ability to participate actively in their studies.

University students perceived web-based learning as a collaborative experience, especially due to the small groups. Small groups strengthened students' involvement and attachment to their studies through social relationships. Web-based group work required students to be engaged and responsible. Each student was responsible, both for his or her own learning, and for the progress of the group. When a group activity was perceived

as successful, its importance was crucial. Some students reported that the groups were self-directed and went beyond the actual studies and that the web-based approach did not challenge the sense of community.

> *"I can study independently and do learning tasks, but working alone can be very tiring. I can draw on the perspectives of other students in my learning, such as in web-based learning. In addition, e-learning is tightly timed: you must work and commit if you want to keep up with others, but on the other hand, you also have to complete your studies on time. The best experience of my e-learning has been in a small group in one course: the learning was fast-paced, and the group was great."*

According to students, the wide range of guidance they received from teachers on web-based learning made it easier to progress and engage with their studies. The guidance and support available when needed gave students confidence in their ability to study and enhanced their overall well-being. The experience of the study community was strong, even though there had never been a physical meeting, which also surprised students. Students appreciated the opportunity to meet new like-minded people of different ages, despite their geographical location. They felt that they had met in person, especially because of the warm and direct interaction with the teachers.

Some of the students complained that their own timetable and goal-oriented progression in their studies discouraged their study methods and made their studies a solitary activity. They had considered various alternative ways of studying, but their own study schedules were a priority. This led to a lack of social dimension in their studies, even though the content of the learning was rewarding and time efficient. Solitary toil was occasionally inspired by small social contacts and the awareness of the existence of peers. Pressure was also eased by the fact that students were aware of the possibility to chat with the teacher and other students on the e-learning platform's discussion board, even if they did not actually do so. Encouraging messages from the teacher to the groups and providing information on current issues also encouraged learning, engaged students, and reinforced learning activities. The students reported that there were variations in the way different teachers interacted in web-based learning environments, and for them the teacher's proactivity was seen as important.

In web-based learning, students had been frustrated by technical problems, especially in the early stages of their studies, which challenged both their well-being and their engagement in their studies. Some perceived technical problems as such a serious threat that they questioned whether they could even study online. For some, the technology caused negative emotional experiences. However, the students persevered in solving the technical problems both independently and by seeking help from their teachers. Friendly and timely support and the concrete help offered with an understanding attitude were, according to the students, crucial to solving the problems. According to the students, potential problem areas, including technical ones, were well anticipated and scheduled online guidance was available. Technical peer support also proved important as the studies progressed, and informal networks were built up. As technical skills improved, the overall perception of web-based learning also became more positive.

### 4.2. The Cognitive Dimension

University students experienced that web-based learning required them to learn new methods of study to progress. For example, the change of learning environment from a physical classroom to an online environment was initially perceived as challenging, but as they learned new methods of learning, the benefits of the online environment became more apparent. The importance of e-books and online lectures was also highlighted, as they were easily accessible and timely. Once the cognitive model of web-based learning was understood, learning became more engaging and effective.

In web-based learning, students' experiences of being able to study independently of time and place were highlighted. The fact that clear structures, instructions, online platforms for discussions and questions, and study timetables had been created for studying,

was also perceived as meaningful and strengthened personal well-being and engagement in studies. Time could be used effectively to promote studies and in-depth learning, and energy was not wasted on perceived unnecessary travel and the acquisition of study materials.

According to the students, the pedagogically well-designed learning activities in the web-based learning were rewarding and were an essential starting point for meaningful learning. For example, the division of a larger learning module into smaller intermediate tasks engaged the students in the study schedule and gave them the appropriate peer pressure. At the same time, the learning processes were constantly ongoing, and the depth of the learning even surprised the students. The deepening of their own learning and the progress of their studies provided a sense of satisfaction, which strengthened their overall well-being, their self-confidence, and engaged them more strongly in their studies.

> *"The great moments are when you have felt the pain of starting and then suddenly you realize how your mind and thoughts have structured and worked on the task after listening to lectures and reading literature. Then it's easy to start putting your thoughts "on paper"."*

University students also emphasized that in web-based academic studies, you also learn about yourself and can learn about your own personal ways of learning. Their own strengths and weaknesses had emerged through experimenting with different methods of learning. In web-based learning, different tasks reflecting on learning have been used to raise awareness of learning styles. For some of the students, it had been meaningful to discover that they knew more than they had thought and that they had been brave enough to try out new ways of learning. Some found that they needed the community and timetable of an online course, while others needed an independent way of studying. In many ways, discovering new things was seen as empowering, which strengthened their confidence in their own agency. For example, it has been an important experience to realize that you do not have to know or find out everything on your own, but that you can find help and guidance if you seek it yourself. In addition, students mentioned that their perception of web-based learning had diversified and become more positive as a whole because of their studies.

> *"I have nothing negative to say, top overall. I gained a lot of new perspectives, learned new things, developed my study strategies and IT skills. Peer learning in an online environment was the right way for me to learn."*

Web-based seminars with live discussion were particularly rewarding, according to students. Collaborative knowledge building was an excellent learning dimension of their studies and was very successful online. However, it was essential that the teacher encouraged discussion, facilitated the situation skillfully and that the group size was small enough. The students also reported that the recordings of the live sessions reinforced their own learning process and retention. In web-based learning, some key skills were even better learned, such as academic writing and citation techniques. They also highlighted the realities of learning. Although learning is independent of time and place, the learning itself requires work, time, and effort. According to the students, the cognitive processes are similar in academic web-based learning, the learning itself always requires effort regardless of the learning environment.

Web-based academic learning had several advantages over traditional learning in a physical learning environment, according to students, such as independence from time and place. They said they did not miss anything essential when studying online. However, university students also had experiences of poor pedagogical solutions in online learning, which were also reflected in the meaningfulness of their learning. In particular, they perceived poor-quality and old lecture recordings from the education provider as undervaluing students. This reflected negatively on their enthusiasm to learn, undermined their confidence in the content relevance of the lecture and led them to critically reflect on the priorities of the education provider. This is an insightful reminder that web-based

solutions need to be made consciously and with the participants' starting points in mind, so that students feel valued, comfortable, and engaged in their studies and learning.

*4.3. The Emotional Dimension*

University students' experience of their own ability to study online varied, and this was reflected in their emotional experience of studying. Students with previous experience of academic study and experience of web-based learning perceived their own preparedness as good. They got off to a fast start, were engaged in their studies, and had a positive emotional experience from the start. Students who were studying online for the first time in academic studies reported that it initially took time and energy to get started with their studies and learning processes. They said that the availability of study guidance and the personal and timely assistance of the teacher were essential. Once they had a sufficient understanding of the study methods and requirements, their sense of their own ability to study was also strengthened. However, they stressed the importance of being proactive and not being discouraged by initial setbacks and feelings of doubt.

> *"I have received a lot of personal guidance online and over the phone. I have got to know the teachers. I have felt good and appreciated. It has been very important that questions are answered as quickly as possible, or if the teacher is busy, they have apologized for the delay. The most important thing, even though the teaching is done remotely and via computers, is that as a student I get the feeling that teaching is between people and that students are considered as individuals."*

A few university students mentioned that the experience of inclusion in web-based learning varies. Sometimes other students take up space in discussions from other participants who want to reflect on perspectives independently first. To address this intensity of participation, the university students hoped that the teacher would come up with pedagogical solutions that consider the possibilities for participation of different types of learners. The students found it frustrating to be marginalized in web-based interactions. However, the diversity of students was reflected, for example, in the fact that some students found learning particularly meaningful when they could complete written learning tasks independently and on their own time. They did not need social relationships in their studies, a need they felt was met in other contexts.

Some of the university students had experience of web-based learning at different universities, i.e., they had extensive experience of web-based pedagogical solutions. They underlined the importance of being treated individually, also online. Previous experience in online learning has shown that students are a mass of people who are financed. This was a negative experience in many ways and had undermined retention. The appreciation and adherence were, according to the students, supported by having access to teachers and a wide range of guidance for their studies. Various acute issues could not wait but had to be addressed within a reasonable and predetermined time frame. According to the students, the sense of knowing the teacher and the teacher knowing the students creates a basis for a pedagogical relationship, including through the online network.

According to the students, the feeling of being able to control their studies, study methods and schedules was meaningful. This enabled them to organize their lives, studies, and work in a holistic way, in a meaningful way. The flexible starting points of web-based learning enabled them to study in a meaningful and personalized way, which strengthened their well-being and engagement in their studies. They hoped that web-based pedagogical approaches could expand and become more widespread with digitalization and also after the pandemic. This would allow more people to have equal access to learning and to update their skills on their own initiative. Online solutions emphasized accountability, freedom, and trust, according to students, who felt that unilateral and coercive practices are no longer relevant in the university context with digitalization.

### 5. Discussion

This study examined how university students' well-being and engagement in web-based learning could be developed during COVID-19. The study focused on the experiences of university students from a case study perspective. The USEI-model [22] was used as a framework for the analysis, with its behavioral, cognitive, and emotional dimensions. First, the study shows that students' behavioral engagement in web-based learning is strong, which differs from previous studies [1,17,18]. Student-centeredness and individual agency are central to the educational approach. Pedagogical solutions in web-based learning enable collaborative activities, interactive small groups, as well as guidance from the teacher to support well-being and engagement [28]. Technical problems in the early stages of learning are a serious threat and need to be carefully anticipated. Second, the cognitive dimension of engagement in web-based learning emphasizes that the students' understanding of the nature and potential of web-based learning increases with experience. The fact that learning is pedagogically well designed and takes place online, respecting the student's own starting points and objectives, enhances the meaningfulness of learning. It is also about learning about oneself and about interacting with other peers. Web-based learning is not a barrier to collaborative knowledge building if the pedagogical solutions are well thought out [4]. Third, the emotional dimension of engagement in web-based learning was related to their own experience of themselves as students. The initial lack of experience in e-learning caused negative emotions, but as the studies progressed, the experience of their own abilities became stronger and their emotions about the studies became more positive. The dimensions of engagement in e-learning overlap and the student experience is holistic. Pedagogical design therefore needs to take these dimensions into consideration, as well as the different starting points of various types of students. It is also important to explore and develop different web-based practices, even if their use is part of everyday activities, as also highlighted by Harju-Luukkainen et al. [2] and Burns et al. [13].

Overall, the USEI model of university students' experiences of web-based learning, wellbeing, and engagement shows that the behavioral, cognitive, and emotional dimensions of university students' academic engagement allow for a holistic approach. The dimensions of engagement in university students' studies cannot be considered in isolation, as the results above show. It is important to consider all of the above dimensions in online academic studies to ensure that students are functioning well and engaged in their studies. In the case of web-based delivery, new pedagogical solutions have been developed rapidly during the pandemic and need to be further developed, with careful attention to students' experiences as one of the drivers for development.

The results convey the positive experiences of university students in terms of web-based learning, well-being, and engagement. The results of this study do not support previous research [18,20]. It is important to highlight that web-based educational practice is at different stages of development. In some institutions, web-based practices have been systematically developed, even before the COVID-19 pandemic. Obviously, the participants in this study have taken part in pedagogically based web-based learning, although academic engagement is multidimensional and, as Marôco et al. [22] state, difficult to define. Education was therefore not in a state of emergency, but digitalization and flexible online practices had been developed on a research basis for a long time, which is reflected in the positive experiences of students. Academic education providers were in very different positions in this respect during and after the pandemic. It is essential to focus on opportunities, but also to recognize weaknesses, such as non-engagement [18] in mass education [12]. This study underlines that online pedagogical solutions were perceived by students as successful, engaging them in their learning, making them feel part of the community, and doing well. Responsible and broadly aware web-based activities in an academic context rely on the best interests of the future society and its members, as Vahed and Rodriguez [14] encourage. Pedagogically high-quality online activities develop at best the resilience and sustainability of societies in the long run. It is important to discuss the results and how they can be interpreted from the perspective of previous studies and of

the working hypotheses. The findings and their implications should be discussed in the broadest context possible. Future research directions may also be highlighted.

## 6. Limitations of This Study

From the point of view of the generalizability of the study, a few limitations must be noted. Only 35 university students participated in the survey. Secondly, qualitative research seeks to describe and understand the phenomenon under study, which means that the size or number of data is not a direct measure of reliability [30]. However, a larger data set could have provided more diverse experiences and perspectives on the phenomenon under study. In addition, it may be considered how the USEI-model was suitable as a framework for analysis. The USEI model has previously been used more in quantitative studies, but it can be used as a theoretical framework for qualitative research and abductive reasoning. Critical reflection by the researchers suggests that the USEI-model brought clarity to the analysis and allowed the perspectives of the model to be examined in relation to university students' experiences of web-based learning and well-being. Any other approach might work, but the starting points chosen in this study are sustainable. Qualitative research does not seek to generalize, and the transferability of results can also be critically assessed [30].

**Author Contributions:** Conceptualization, M.M. (Minna Maunula), M.M. (Minna Maunumäki), J.M. and H.H.-L.; methodology, M.M. (Minna Maunula); writing—original draft preparation, M.M. (Minna Maunula), M.M. (Minna Maunumäki), J.M. and H.H.-L.; writing—review and editing, M.M. (Minna Maunula), M.M. (Minna Maunumäki) and H. H-L.; visualization, M.M. (Minna Maunula); supervision, M.M. (Minna Maunula), M.M. (Minna Maunumäki), J.M. and H.H.-L.; project administration, M.M. (Minna Maunula). All authors have read and agreed to the published version of the manuscript.

**Funding:** This research received no external funding.

**Institutional Review Board Statement:** The study was conducted in accordance with the Declaration of Helsinki and approved by the institutional review board of TENK.

**Informed Consent Statement:** Informed consent was obtained from all subjects involved in the study.

**Data Availability Statement:** Not applicable.

**Conflicts of Interest:** The authors declare no conflict of interest.

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
