# Peer review of "Developing Students Well-Being and Engagement in Higher Education during COVID-19—A Case Study of Web-Based Learning in Finland"

_sustainability, doi:10.3390/su15043838_

Round 1

Reviewer 1 Report

The article is well-conceived and deserves to be published. However, some issues must be put in perspective and should be highlighted and punctuated. In the first sentence, “The COVID-19 disrupted normal routines of societies” is written, however, what is normal in a society? I would suggest the sentence be removed.

In the same way, “too many sectors of society” is more precise, describing which sectors of society. For this matter, I would suggest attaching two theories about the pandemic, education, and the digital environment. Although, both are not specialists in Finnish education João de Castro Rocha writes about the technique of the Brazilian state, the shrinking of public policies during the pandemic, and the cultural war element from the digital universe in his book João Cezar de Castro Rocha, Guerra cultural e retórica do ódio: crônicas de um Brasil pós-político, Goiânia: Caminhos, 2021.

Another theory held by a Brazilianist is the one that connects the use of hatred during the pandemic from religious actors to disqualify plural education alongside digital platforms encouraging the expansion of digital engagement, which was written by Fábio Py in his “The Current Political Path of an Ultra-Catholic Agent of Brazilian Christofacism Father Paulo Ricardo”, International Journal of Latin American Religions, v.5, 2021. Both theories had helped to design a framework of experiences that link the pandemic, education, and the digital realm. That being said, the highlight is that within other geographies there are different forms of “discomfort” in the face of the expansion of digital in the pandemic in peripheral countries.

At the same time, I miss a more direct discussion about the “emotional” in social experiences when it unites “social experiences and meanings” and it is worth the deepening gave by Le Breton, in his Antropologia dos sentidos, Petrópolis: Vozes, 2016.

Hence I believe the article is well developed in terms of method and results, and therefore, these indications and annexes right at the beginning will help you in showing theories and social debates, especially concerning the pandemic and its social impacts. The article deserves to be published as long as it assumes all the elements indicated in the analysis.

Reviewer 2 Report

Dear Authors,

After reviewing the article, I will proceed to list some comments so that you can improve its quality, since I consider that it can be published with minor changes.

First, on the theoretical framework:

-          I consider that the theoretical framework is high quality.  It fits the topic and relevant sources are reviewed taking into account multiple related variables.

-          I understand it is necessary to include the information on the research question and the explanation on the justification of the relevance of the research at the end of the theoretical framework, which is not the section where it is located.

Second, on the materials and methods:

-          I understand it is necessary to subdivide the method and materials section into different subsections. Thus, I recommend separating the information related to the participants (it would not be superfluous to expand the information about them), the data collection procedure and the content analysis. I understand that it would also be necessary to explain much more about the type of techniques used to collect the data.

In third place, in relation to results:

-          Since data triangulation techniques are mentioned, which undoubtedly increases the validation and reliability of the study. I consider it necessary to provide some visual representation of them. I recommend including networks that illustrate the density of data and the relationship between them.

Fourth, regarding the discussion and conclusions

-          I recommend that the limitations section of the study be placed at the end of the article before the authors' contributions.

-          I recommend including a section on general conclusions at the end of the discussion.

-          I also recommend reviewing the use of ( ; ) and ( , ) throughout the document when including citation numbers. The use of ( ; ) to separate numbers should be consistent throughout the document.

I hope you will be able to attend to these considerations, which are always aimed at improvement.

With my best wishes.

Round 2

Reviewer 1 Report

I am satisfied with the changes to the article.
Thanks